# Anti-Inflammatory Effects of Resveratrol on Human Retinal Pigment Cells and a Myopia Animal Model

**Yu-An Hsu** [1,†], **Chih-Sheng Chen** [2,3,†], **Yao-Chien Wang** [4], **En-Shyh Lin** [5], **Ching-Yao Chang** [6], **Jamie Jiin-Yi Chen** [7], **Ming-Yen Wu** [7], **Hui-Ju Lin** [1,7,*] **and Lei Wan** [1,6,8,*]

1   School of Chinese Medicine, China Medical University, Taichung 404, Taiwan; annhsu007@gmail.com
2   Department of Food Nutrition and Health Biotechnology, Asia University, Taichung 413, Taiwan; pluto915@mail2000.com.tw
3   Division of Chinese Medicine, Asia University Hospital, Taichung 413, Taiwan
4   Department of Emergency Medicine, Taichung Tzu Chi Hospital, Taichung 427, Taiwan; yuan5841107@hotmail.com
5   Department of Beauty Science, National Taichung University of Science and Technology, Taichung 404, Taiwan; eslin@nutc.edu.tw
6   Department of Medical Laboratory Science and Biotechnology, Asia University, Taichung 413, Taiwan; cychang@asia.edu.tw
7   Eye Center, China Medical University Hospital, Taichung 404, Taiwan; sealily007@yahoo.com.tw (J.J.-Y.C.); uk11234@hotmail.com (M.-Y.W.)
8   Department of Obstetrics and Gynecology, China Medical University Hospital, Taichung 404, Taiwan
*   Correspondence: d2396@mail.cmuh.org.tw (H.-J.L.); leiwan@cmu.edu.tw (L.W.); Tel.: +886-4-22053366 (H.-J.L. & L.W.)
†   These authors contributed equally to this work.

**Abstract:** Resveratrol is a key component of red wine and other grape products. Recent studies have characterized resveratrol as a polyphenol, and shown its beneficial effects on cancer, metabolism, and infection. This study aimed to obtain insights into the biological effects of resveratrol on myopia. To this end, we examined its anti-inflammatory influence on human retinal pigment epithelium cells and in a monocular form deprivation (MFD)-induced animal model of myopia. In MFD-induced myopia, resveratrol increased collagen I level and reduced the expression levels of matrix metalloproteinase (MMP)2, transforming growth factor (TGF)-β, and nuclear factor (NF)-κB expression levels. It also suppressed the levels of tumor necrosis factor (TNF)-α, interleukin (IL)-6, and IL-1β. Resveratrol exhibited no significant cytotoxicity in ARPE-19 cells. Downregulation of inflammatory cytokine production, and inhibition of AKT, c-Raf, Stat3, and NFκB phosphorylation were observed in ARPE-19 cells that were treated with resveratrol. In conclusion, the findings suggest that resveratrol inhibits inflammatory effects by blocking the relevant signaling pathways, to ameliorate myopia development. This may make it a natural candidate for drug development for myopia.

**Keywords:** resveratrol; anti-inflammation; myopia; MFD; RPE cells

## 1. Introduction

Myopia is a common type of refractive error and represents a worldwide public health problem, with social and economic implications [1]. Myopia occurs due to eye elongation, associated with decreased connective tissue synthesis and increased collagen I degradation, which remodels the composition of the sclera [2]. Recent studies have indicated that the retina, photoreceptors, and retinal pigment epithelium play important roles in the regulation of scleral tissue remodeling, by signal activation for ocular growth and axial length [3]. In myopic eyes, various molecules have been shown to be involved in myopia development, including transforming growth factor (TGF)-β, matrix metalloproteinase (MMP)-2, and collagen I. TGF-β is expressed in ocular tissues and is known to remodel the scleral extracellular matrix (ECM) [4–6], resulting in reduced collagen production. The

collagen disruption is elevated through the activation of nuclear factor (NF)-κB, increasing matrix metalloproteinase 2 (MMP2) expression and reducing the activity of the tissue inhibitor of MMP2 [7]. MMP2 has also been shown to be upregulated in the sclera of chicks and tree shrews, by form deprivation-induced myopia [8]. Recent clinical and experimental studies provided evidence showing that atropine induced the downregulation of c-Fos, NF-κB, interleukin (IL)-6, and tumor necrosis factor (TNF)-α expression in hamsters with monocular form deprivation (MFD)-induced myopia [9]. These findings indicate that inflammation plays a crucial role in the development of myopia.

Resveratrol (3,4′,5 trihydroxy-stilbene) is a naturally occurring phytoalexin, in plants such as grapes, that provides resistance against microbial or fungal infections, or stressful stimuli. Resveratrol can modulate various intracellular enzymes, including kinases, lipoxygenases, cyclooxygenases, and scavengers of free radicals [10]. Recent studies have shown that resveratrol has various therapeutic effects in cancers, diabetes, and cardiovascular disease [11]. The effects of resveratrol are indicated by decreased serum levels of inflammatory cytokines such as TNF-α, IL-1β, and IL-6, and suppression of the NF-κB pathway for cardiovascular diseases in rats [10,12]. Resveratrol inhibited inflammasome activation by the downregulation of NFκB, and P38 mitogen-activated protein kinase (MAPK) expression, while upregulating sirtuin 1 (SIRT1) expression to protect from vascular injury [13,14]. It also suppressed the invasion activity of the human hepatocellular carcinoma cell, through prohibiting TNF-α-mediated MMP9 expression. Resveratrol exhibited anti-obesity effects via the downregulation of adipogenic genes (PPARγ, C/EBPα, FAS, aP2) expression, and attenuated the pro-inflammatory cytokines (TNF-α and IL-6) expression [15]. Moreover, resveratrol has been shown to be an activator of Sir2, which is a conserved deacetylase for replicative lifespan in aging research [16,17]. These effects were linked with the anti-inflammatory activity of resveratrol [18–20].

A recent study has demonstrated the role of inflammation in myopia progression [9]. Furthermore, studies have indicated that the anti-inflammatory activity of resveratrol suppresses inflammation in multiple diseases, as well as cancer [21]. Nevertheless, the effects of resveratrol on myopic eyes are still not well understood. In this study, we investigated the effects of resveratrol on the development of MFD-induced myopia in a hamster model. Accordingly, the modulatory mechanism of resveratrol was also tested in human retinal pigment epithelium cells.

## 2. Material and Methods

### 2.1. Cell Culture

Human retinal pigment epithelial cells (ARPE-19 cells) were purchased from Bioresource Collection and Research Center, HsinChu, Taiwan (BCRC; BCRC-60383). The cells were cultured in Dulbecco's modified Eagle's medium (DMEM; Thermo Fisher Scientific, Waltham, MA, USA) containing 10% fetal bovine serum (FBS), 100 μg/mL streptomycin sulfate, and 100 U/mL penicillin G sodium at 37 °C and 5% $CO_2$.

### 2.2. Cell Viability Test

Cell viability was determined using the MTS assay. ARPE-19 cells ($5 \times 10^3$ cells/well) were seeded in 96-well plates. Resveratrol (10 μg/mL) was purchased from Shanghai Yuanye Bio-Technology and immersed in DMSO. Media containing different concentrations (0 μg/mL, 4 μg/mL, 2 μg/mL, 1 μg/mL, 0.5 μg/mL, 0.25 μg/mL, 0.125 μg/mL, 0.0625 μg/mL, and 0.03125 μg/mL) of the resveratrol extract and 0.5% DMSO were added and incubated for 24 h. Every dosage was treated using six wells for replicates. Subsequently, a 20 μL mixture of MTS/PMS ((3-(4,5-dimethylthiazol-2-yl)-5-(3-carboxymethoxyphenyl)-2-(4-sulfophenyl)-2*H*-tetrazolium inner salt) (Promega, Madison, WI, USA) 2 mL/phenazine methosulfate 100μL (Sigma-Aldrich, St. Louis, MO, USA) was added and incubated for an additional 2 h. The absorbance was read at 490 nm in the microplate reader SpectraMax® ABS plus (Molecular Devices, San Jose, CA, USA).

### 2.3. Enzyme-Linked Immunosorbent Assay (ELISA)

Approximately $3 \times 10^5$ cells/well in 2 mL of culture medium were seeded into 6-well plates, incubated overnight, and stimulated with 5 ng of a combination of the cytokines TNF-$\alpha$, IL-6, and IL-1$\beta$ (Peprotech, Rock Hill, SC, USA) for 10 min. Cells treated with the cytokine combination were subsequently treated with 2 µg/mL, 1 µg/mL, 0.5 µg/mL, 0.25 µg/mL, or 0.125 µg/mL of the resveratrol extract for 2 h. All samples were stored at $-20\,^\circ$C before further assays. The concentration of IL-6, monocyte chemoattractant protein-1 (MCP-1), and IL-8 in culture supernatants was measured using ELISA kits (Thermo Fisher Scientific, Waltham, MA, USA) according to the manufacturer's instructions. The value of each treatment was determined from triplicates and replicated in three independent experiments. The absorbance was read at 450 nm with the microplate reader SpectraMax® ABS plus (Molecular Devices, San Jose, CA, USA).

### 2.4. Animals and MFD Induction

Three-week-old golden Syrian hamsters weighing 80–90 g were used for the experiments. The mice were obtained from Taiwan National Laboratory Animal Center and National Applied Research Laboratories (NARLabs, Taipei, Taiwan), and housed according to the principles of laboratory animal care. The procedures for animal care and handling were approved by the animal committee of China Medical University (permit number: 103–221 NH) and were in accordance with the ARRIVE guidelines for the use of animals in ophthalmic and vision research.

The myopia in eyes exposed to MFD is caused by disruption of the refractive development [22]. The axial lengths were measured by A-scan ultrasonography (PacScan 300 plus, New Hyde Park, NY, USA) before eyelid fusion. MFD was performed in the right eye for 21 days under anesthesia induced by the intraperitoneal injection of 50~80 mg/kg of Zoletil 50 (Virbac, Nice, France), depending on the animals' response.

The animals were randomly allocated into three experimental groups (control, 1% atropine and 100 mM resveratrol). Each group consisted of 10 animals. Both the left eye, which served as an internal control, and the right eye, received daily application of drugs (1% atropine or 100 mM resveratrol) or balanced salt solution (BSS) in the treatment and control groups, respectively. Atropine, which was used as a positive control, has been indicated to inhibit myopia progression by downregulating inflammation in MFD eyes [9]. Administered in the indicated dose of 20 µL twice daily with 8 h between the administrations for 21 days. At the end of study, the axial lengths were measured and 10 different records were averaged for the axial length. Animal eyes were collected for immunohistochemistry and immunofluorescence examinations after euthanization through $CO_2$ asphyxiation.

### 2.5. Preparation of Resveratrol Solution

Resveratrol solution (100 mM) was dissolved with 2-hydroxypropyl-$\beta$-cyclodextrin (HP-$\beta$-CD) in the following steps. First, hydroxypropyl-$\beta$-cyclodextrin was solubilized water and resveratrol was dissolved in 100% ethanol. Next, resveratrol was added drop-by-drop with continuous stirring into the hydroxypropyl-$\beta$-cyclodextrin solution at a molar ratio of 1:2. The mixture was then lyophilized and dissolved in balanced salt solution (BSS).

### 2.6. Immunohistochemical and Immunofluorescence Analyses

Eye tissues were dissected and fixed in 10% formalin solution for 1 day. After dehydration in ethanol, the tissue was embedded in paraffin and cut into 8-µm-thick sagittal sections and collected on glass slides. Sections were deparaffinized, and antigen retrieval was performed by boiling the slides in citrate buffer (pH 6.0), immersed in hydrogen peroxide for 5 min, and blocked by PBS containing 5% normal rabbit serum for 1 h at room temperature. Sections were then stained with primary antibodies against NF-$\kappa$B (Cell signaling, Beverly, MA, USA), collagen I (Novus Biologicals, Centennial, CO, USA), TGF-$\beta$, MMP2, TNF-$\alpha$ (Abcam, Boston, MA, USA), IL-6, and IL-1$\beta$ (Abcam, Boston, MA,

USA) overnight at 4 °C. Then the Novolink polymer detection systems (Leica, Newcastle, UK) were used according to the manufacturer's instructions, to visualize immunoreactivity. The slides were then observed under a microscope (Olympus, Tokyo, Japan) at 20× magnification. For immunofluorescence staining, the primary antibody of TNF-α was also used and subsequently incubated with anti-rabbit secondary antibody labeled with Cy3 (Jackson ImmunoResearch Laboratory, West Grove, PA, USA) and DAPI for DNA staining. Sections were imaged using Leica DMi8 microscope with MetaVue software, version 7.8.8.0 (Molecular Devices, San Jose, CA, USA). The images were processed using ImageJ 1.49 software (NIH, Bethesda, MD, USA) for quantification of the relative intensity of each antibody. The relative values were represented as fold changes as compared to control group.

### 2.7. Western Blotting Analysis

ARPE-19 cells were washed twice with cold phosphate-buffered saline and lysed with RIPA buffer supplemented with a protease and phosphoprotease inhibitor mixture (Roche). The protein concentrations of the cell lysis extracts were measured using Bradford protein assay (Bio-Rad, Hercules, CA, USA) and equalized with extraction reagent. Equal amounts (20 μg) of the extracts were loaded and fractionated by sodium dodecyl sulfate-polyacrylamide gel electrophoresis (SDS-PAGE), transferred onto 0.45 μm polyvinylidene fluoride membranes (PVDF; Millipore, Billerica, MA, USA) using a transfer apparatus, according to the manufacturer's protocols (Bio-Rad, Hercules, CA, USA). After blocking with 5% nonfat milk in TBST for 1 h, the membranes were incubated with appropriate primary antibodies (1:1000 dilution; AKT, phospho-AKT, c-Raf, phospho-c-Raf, stat3, phospho-stat3, NFkB, phosphor- NFkB and beta actin: Cell Signaling Technologies, Danvers, MA, USA) overnight at 4 °C. The membranes were washed thrice in PBST and incubated with horseradish peroxidase-conjugated anti-mouse or anti-rabbit antibodies (1:5000~10,000 dilution; Cell Signaling) at room temperature for 1 h. The membranes were then washed thrice in PBST and developed using ECL system (Millipore). The reaction was visualized by chemiluminescence using an ImageQuant LAS4000 mini (GE Healthcare, Little Chalfont, UK) system. Immunoblots were quantified using the ImageJ 1.49 software (NIH, Bethesda, MD, USA).

### 2.8. Statistical Analysis

All data are presented as mean ± standard deviation (SD) with at least three independent experiments. Differences between the two experimental groups were analyzed statistically with the unpaired independent *t*-test. One-way ANOVA analysis was performed to compare three groups using GraphPad prism software (version 7, San Diego, CA, USA). Differences in means were considered significant for *p* values < 0.05.

## 3. Results

### 3.1. Resveratrol Inhibits the Progression of Myopia by Suppressing Inflammation in the MFD Animal Model

We first determined whether resveratrol influences the development of myopia, by decreasing the levels of myopia-related tissue remodeling proteins and inflammation effects in the MFD animal model. In the present study, resveratrol and 1% atropine were applied to the right MFD eyes and the left eyes without MFD, and the axial length was measured after 21 days. As shown in Figure 1A, the changes in the axial length in the control, 1% atropine, and resveratrol groups were $0.405 \pm 0.024$, $0.334 \pm 0.016$, and $0.329 \pm 0.0128$, respectively (ANOVA *p* < 0.05). The ability of resveratrol to suppress the inflammatory activity was also investigated, by immunohistochemical analyses in the MFD eyes. In comparison with the control treatment, the resveratrol-treated MFD eyes showed reduced expression of NFκB (Figure 1B), increased collagen I expression, and inhibition of TGF-β and MMP2 expression (Figure 1C), and significant suppression of the TNFα, IL-6 and IL-1β expression induced by MFD (Figure 1D). The results of the immunofluorescence analyses, in comparison with the

control treatment, showed that resveratrol significantly inhibited the expression of TNF-$\alpha$ in the retinal pigment epithelium layer, similarly to the effects of atropine (Figure 1E). These data suggest that resveratrol has similar regulatory effects to atropine, and that it inhibited myopia progression in MFD eyes by modulating the changes in the tissue-remodeling proteins and the inflammatory effects.

### 3.2. Resveratrol Shows No Significant Effect on the Viability of ARPE-19 Cells

Next, we tried to determine the biological effects of resveratrol on retinal pigment epithelial (RPE) cells. The retinal pigment epithelium (RPE) has been reported to be correlated with myopia development [23]. Therefore, the effects of resveratrol on cell viability in the human retinal pigment epithelium cell line ARPE-19 were assessed by the MTS assay. The cells were treated with the indicated concentration of resveratrol (4, 2, 1, 0.5, 0.25, 0.125, 0.0625, 0.03125 $\mu$g/mL) for 24 h. As shown in Figure 2, resveratrol treatment resulted in approximately 100% viability at concentrations up to 2 $\mu$g/mL, and resveratrol treatment at 4 $\mu$g/mL only exhibited about 7% inhibition of ARPE-19 cell viability. These findings indicated that resveratrol showed no significant cytotoxicity for ARPE-19 cells at concentrations of 0.03125–4 $\mu$g/mL.

### 3.3. Resveratrol Inhibits the Inflammatory Effects of ARPE-19 Cells

ARPE-19 cells were incubated with inflammatory cytokines (TNF-$\alpha$, IL-6, IL-1$\beta$, or combinations of these cytokines), 10 min prior to the treatment with resveratrol, for 2 h, to examine the effects of resveratrol on the inflammatory cytokine-induced production of IL-6, MCP-1, and IL-8, which was determined by ELISA analysis. Pre-treatment with a combination of TNF-$\alpha$, IL-6 and IL-1$\beta$ led to the highest expression levels of IL-6, MCP-1, and IL-8. However, incubation with the indicated concentrations of resveratrol resulted in significant inhibition of IL-6, MCP-1, and IL-8 expression, in a dose-dependent manner (Figure 3A–C). These results indicate that resveratrol has an inhibitory effect on the combination of TNF-$\alpha$-, IL-6-, and IL-1$\beta$-induced inflammatory reaction.

### 3.4. Resveratrol Reduces the Activation of Inflammatory Cytokine-Induced AKT, c-Raf, Stat3, and NF$\kappa$B in ARPE-19 Cells

In order to study the effect of resveratrol on the PI3K–AKT, MAPK, and NF$\kappa$B pathways, involved in myopia progression, ARPE-19 cells were incubated with TNF-$\alpha$, IL-6, and IL-1$\beta$, alone and with combinations of these cytokines, for 10 min prior to the 10 min treatment with resveratrol. This allowed us to examine the effects of resveratrol on inflammatory cytokine-induced signal factors. AKT, c-Raf, stat3 and NF$\kappa$B phosphorylation in the cell lysates was examined by Western blot analysis, using anti-AKT, anti-c-Raf, anti-stat3, and anti-NF$\kappa$B, respectively. Treatment with resveratrol led to significant inhibition of AKT, c-Raf, and NF$\kappa$B phosphorylation (Figure 4A–D), compared with the combination of TNF-$\alpha$, IL-6 and IL-1$\beta$. AKT, c-Raf, and NF$\kappa$B activation can prompt TNF-$\alpha$ and IL-6 expression. TNF-$\alpha$ and IL-6 may act a feedback loop to activate the Stat3, AKT, and NF$\kappa$B signaling pathway, through binding to their receptors. On the ARPE-19 cells, treatment with resveratrol also inhibited Stat3 phosphorylation (Figure 4C,D). Thus, the results indicated that resveratrol inhibited the activation of inflammation-induced signal factors, and suggested that resveratrol can suppress the inducers of myopia-related molecule expression.

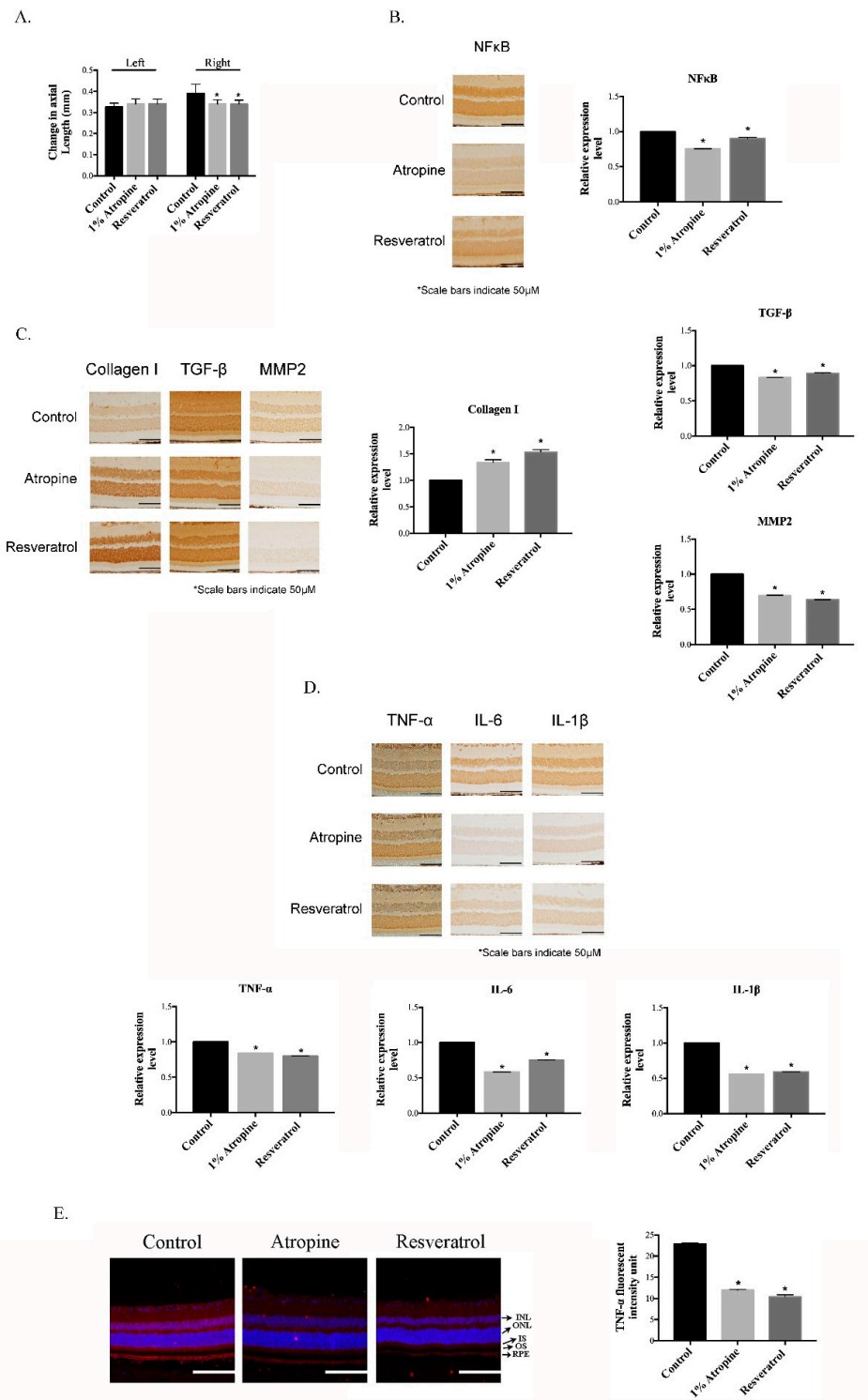

**Figure 1.** Effect of resveratrol on myopia progression. (**A**) The axial length was determined as the difference in diopter measurements taken after and before MFD. * $p < 0.05$ indicates significant differences from the control. Quantification of immunohistochemical staining for NFκB (**B**), collagen I, TGF-β, and MMP2 (**C**), and TNF-α, IL-6, and IL-1β (**D**). (**E**) Immunofluorescence staining with TNF-α and quantification analysis of the RPE layer by Image J software. Results are shown as mean ± SD; The scale bar indicates 50 μm, 20× magnification for IHC stain images, 100 μm, 20× magnification for IF stain images * $p < 0.05$, $n = 3$ for each.

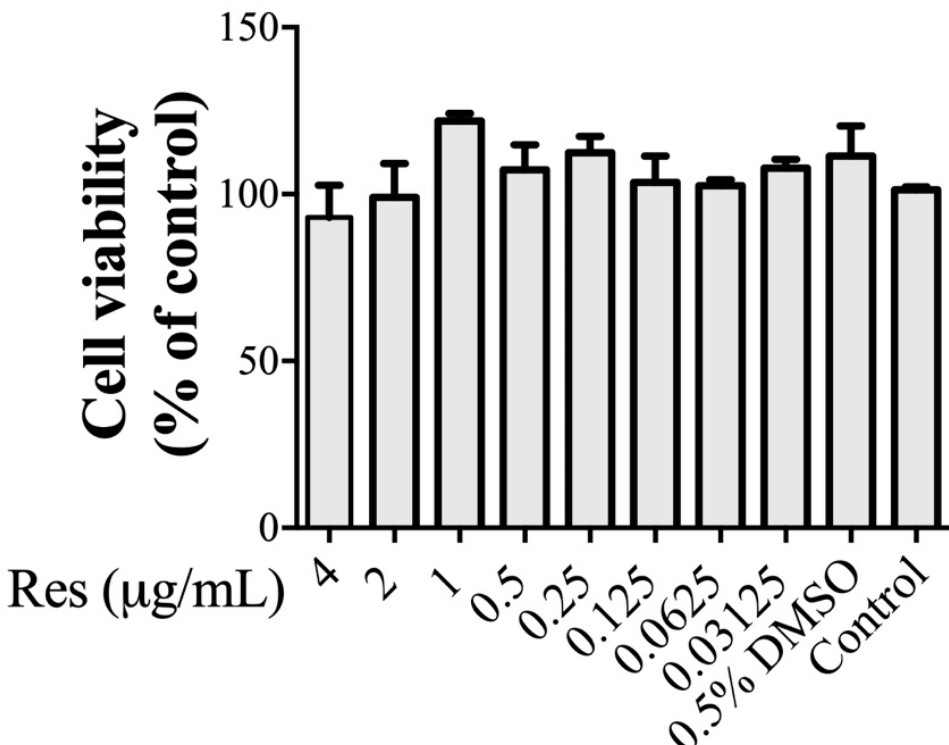

**Figure 2.** Effect of resveratrol on cell viability in ARPE-19 cells. Cells were treated with resveratrol at 4, 2, 1, 0.5, 0.25, 0.125, 0.0625, 0.03125 μg/mL for 24 h. At the end of the respective treatments, MTS assays were performed as described in the materials and methods. The ratios are expressed as the percentage of viable cells normalized to the untreated control. The values of the MTS assay are mean ± SE of six replicates in each treatment.

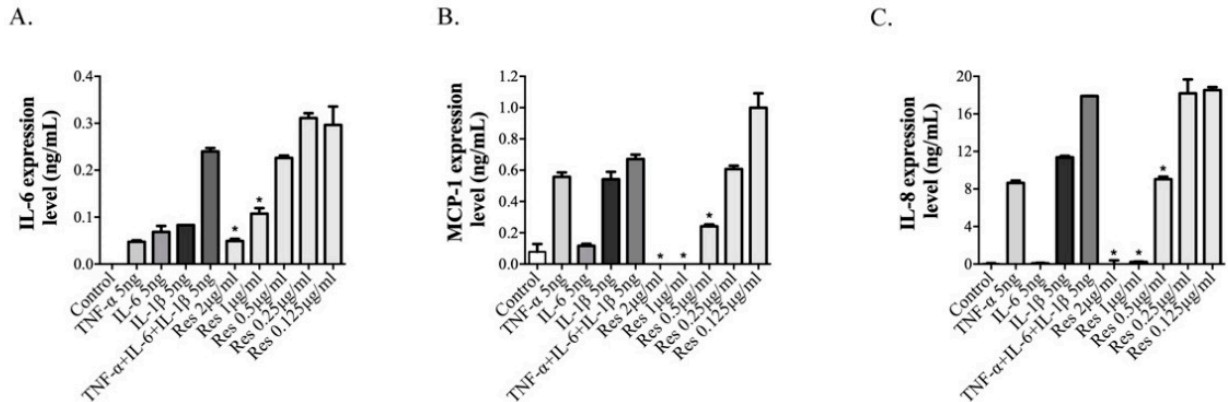

**Figure 3.** Effect of resveratrol on inflammatory cytokine expression in pro-inflammatory cytokine-stimulated ARPE-19 cells. Cells were pretreated with pro-inflammatory agents (TNF-α, IL-6, and IL-1β alone and in combinations) for 10 min and subsequently treated with 0.125–2 μg/mL of resveratrol for 2 h. (**A**) IL-6, (**B**) MCP-1, and (**C**) IL-8 levels were measured by ELISA. The values represent the mean ± SD of triplicate determinations from three independent experiments. * $p < 0.05$ indicates significant differences between combinations of pro-inflammatory agents and resveratrol.

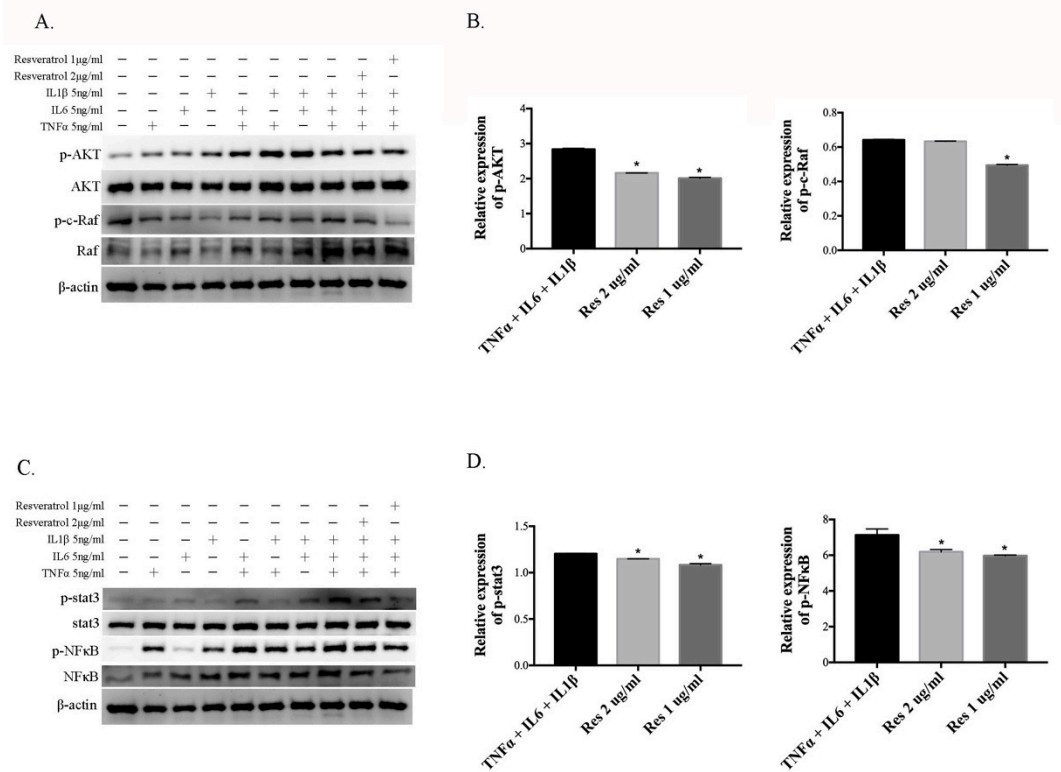

**Figure 4.** Effects of resveratrol on transcription factor expression in pro-inflammatory cytokine-stimulated ARPE-19 cells. Cells were pretreated with pro-inflammatory agents (TNFα, IL-6, IL-1β alone and in combination) for 10 min and subsequently treated with 2 and 1 μg/mL of resveratrol for 10 min. (**A**) p-AKT, AKT, p-c-Raf, and c-Raf levels were measured by immunoblot. The β-actin immunosignal was used as loading control. (**B**) The AKT and c-Raf relative fold changes in expression levels were normalized with β-actin and the respective un-phosphorylated forms by ImageJ. (**C**) Western blot analysis for p-stat3, stat3, p-NFκB, and NFκB. (**D**) The stat3 and NFκB relative fold changes in expression levels were normalized with β-actin and respective un-phosphorylation form by ImageJ. Data are shown as the mean ± SD of triplicate determinations from three independent experiments. * $p < 0.05$ indicates significant differences between the combination of pro-inflammatory agents and resveratrol.

## 4. Discussion

The present study has established that the axial changes of eye and inflammatory factor changes in the retina and RPE, during induced myopia, in response to resveratrol treatment. In Figure 1A, the findings demonstrated a decrease in axial length in myopia development, under resveratrol treatment. Myopia-related tissue remodeling proteins and inflammatory cytokines (NFκB, TGF-β, MMP2, TNFα, IL-6 and IL-1β) expression presented a decrease in resveratrol-treated MFD eyes. An increase in collagen I expression, in resveratrol-treated MFD eyes, was also observed (Figure 1B–E). These findings support the hypothesis that resveratrol displays a similar regulation pattern as atropine.

The biological effects of resveratrol on ARPE-19 cells were investigated, showing no significant cytotoxicity at concentrations of 0.03125–4 μg/mL (Figure 2). Resveratrol has been demonstrated to enhance cell proliferation in low doses (0.1–1 μg/mL) [24]. Previous findings are similar to ours, where a higher percentage of viable cells was found at 0.03125–1 μg/mL than those in the control. This result suggested that a high and low dose of resveratrol both have different cellular modulation. Therefore, we choose 2 and 1 μg/mL concentration for further experiments, to evaluate the effect of resveratrol.

Resveratrol is isolated from the roots of *Polygonum cuspidatum*, which is used for the treatment of inflammation, carcinogenesis, and cardiovascular disease in traditional medicine [25]. Resveratrol was initially characterized as a phytoalexin that conferred the plant with resistance against microbial or fungal infections, and stressful stimuli such as

injury or ultraviolet radiation [26]. Although it is found in various food sources, it is better known as a constituent of grapes and wines. Romero-Pérez et al. demonstrated that the resveratrol content in red wines is higher than that in white wines, because the skin is removed earlier in the white wine production process [27].

Resveratrol has shown anti-inflammatory, antioxidant, and anti-neoplastic effects, and has also shown therapeutic properties in various diseases. In assessments of metabolic derangements, some studies suggested that resveratrol increased the expression and activity of the protein deacetylase enzyme-silent information regulator 2/sirtuin-1 (SIRT1), which is a central molecule that modulated the body's response to diet and exercise [28–30]. In mice fed a high-caloric diet, treatment with resveratrol also increased insulin sensitivity and reduced insulin-like growth factor-1 (IGF-1) levels, to improve the lifespan [15,31,32]. Additionally, the reported effects of resveratrol on metabolic syndrome include fat mass reduction, body weight loss, and reductions in the plasma levels of triglyceride, TNF-$\alpha$ and MCP-1. In cardiovascular diseases, the anti-inflammatory effects of resveratrol include inhibition of intracellular adhesion molecule 1 (ICAM-1), and induction of nitric oxide synthase (iNOS) and IL-1$\beta$ mRNA expression in coronary arterial endothelial cells [10]. Cardiovascular diseases can result in heart failure, and one study showed that macrophage and mast-cell infiltration, by transverse aortic constriction surgery-induced heart failure, were significantly attenuated by oral administration of resveratrol in C57BL/6 mice [33].

In respiratory disease, considering asthma as an example of a disease that mainly presented with lower respiratory tract chronic inflammation, increased airway hyperresponsiveness (AHR), and mucus production, resveratrol treatment induced reductions in IL-4, IL-5, TNF-$\alpha$, and TGF-$\beta$ cytokine levels in an OVA-induced asthma model [34]. Moreover, the anti-inflammatory effects of resveratrol included suppression of AHR and reduction in the infiltration of eosinophil cells into bronchoalveolar lavage fluid (BALF) and lung tissue [35,36]. In addition, several studies have indicated that resveratrol reduced the production of NO and TNF-$\alpha$, by attenuation of the LPS-induced phosphorylation of p38–MAPK and degradation of I$\kappa$B inhibitors in microglial cells [37]. Resveratrol inhibited macrophage infiltration into the RPE–choroid, and prevented the CNV-induced AMPK decrease and NF$\kappa$B increase in the RPE–choroid complex, to avoid CNV development [38].

Resveratrol also shows in vitro and in vivo anti-inflammatory activity against A$\beta$-triggered microglial activation, via downregulation of the TLR4/NF-$\kappa$B/STAT signaling cascade [20,39]. The findings described above illustrate how the beneficial effects of resveratrol are largely associated with its anti-inflammatory activity, in decreasing the serum levels of inflammatory factors such as TNF-$\alpha$, IL-1$\beta$, and IL-6, and suppressing the p38–MAPK and NF–$\kappa$B pathways [40]. Nevertheless, current research on the benefits of resveratrol for eye health have proven to be inconclusive. A recent study showed that resveratrol prevents retinal mitochondrial activation, by modulating SIRT-1 and Ku70 upregulation, and the resulting BAX into the mitochondria has been prohibited [41].

Resveratrol combined with bevacizumab reverses age-related macular degeneration (AMD) and proliferative diabetic retinopathy (PDR) caused by neo-vascular changes in the retina, which were also observed through notch signaling involvement [42,43].

With advancements in technology, the prevalence of myopia has rapidly increased in recent decades. Myopia is the most common type of refractive error and results in blurred distance vision, because images of distant objects are focused in front of the retina. It is caused by dynamic changes in the scleral tissue that influence ocular development [44]. Previous studies have shown that the levels of the major structural protein, collagen I, were reduced in the sclera [45]. The loss of collagen I increased the expression of MMP2, resulting in changes in the composition and ductility of the sclera. MMP2 expression has been shown to be upregulated by form deprivation in the sclera of chicks and tree shrews [8,46]. TGF-$\beta$ modulates the MMP2 expression level via NF-$\kappa$B activation, to regulate various inflammatory cytokines in fibroblasts [7]. The inflammatory cytokines, such as MCP-1, IL-6, IL-8, and TNF-$\alpha$, showed higher expression levels in allergic eyes,

which were in an inflammation stage. Therefore, the relationship between inflammation and myopia has been investigated [47].

In this study, the hamsters in which myopia was induced by the MFD model, showed higher MMP2 and TGF-β levels and a lower collagen I level, similar to the findings obtained in previous studies on myopic eyes. However, treatment with resveratrol reduced the levels of MMP2 and TGF-β, and rescued collagen I expression. Moreover, resveratrol downregulated the expression levels of IL-6, IL-8, and TNF-α, by inhibiting NF-κB activation in the MFD-induced myopic animal model. Recent studies have indicated that retina-specific photoreceptors and RPE play an important role in eye growth regulation and axial length changes, by delivering signal transduction for tissue remodeling of the sclera [44,48]. The RPE is a monolayer of highly specialized pigmented cells that plays a critical role in photopigment regeneration by the phagocytosis of photoreceptor outer segments, facilitating the uptake and recycling of retina, to maintain its photoreceptor functions [49]. In our investigation, resveratrol efficiently showed inflammatory inhibition of the RPE layer of MFD-induced myopic eyes (Figure 1). Furthermore, resveratrol also exhibited notable inhibition of the pro-inflammatory cytokines (TNF-α, IL-6 and IL-1β) that are induced by inflammation in ARPE-19 cells (Figure 3). In the RPE cells, we also demonstrated that resveratrol improved efficacy via AKT, C-Raf, and NF-κB signal transduction, to influence the anti-inflammatory effect (Figure 4).

Taken together, the findings of the present study suggest that the anti-inflammatory activity of resveratrol prevents MFD-induced myopia in a hamster animal model and RPE cells. We demonstrated that resveratrol can inhibit the MFD-induced expression of inflammatory factors such as TNF-α, IL-1β, IL-6, and TGF-β via suppression of the AKT, c-Raf, NF-κB and STAT3 pathways. Therefore, these results imply that resveratrol could contribute to the treatment of a variety of inflammatory diseases, including myopia.

**Author Contributions:** L.W. and H.-J.L. conceived and designed the experiments. Y.-A.H., C.-S.C. performed the experiments. C.-Y.C., J.J.-Y.C. and M.-Y.W. analyzed and interpreted the data. Y.-C.W. and E.-S.L. contributed reagents, materials, analysis tools or data. Y.-A.H., C.-S.C. wrote the paper. All authors have read and agreed to the published version of the manuscript.

**Funding:** This study was supported in part by the Ministry of Science and Technology, Taiwan, R.O.C. (MOST107-2320-B-039-049-MY3, MOST105-2628-B-039-0008-MY3 and MOST108-2314-B-039-048-MY3), China Medical University Hospital, Taichung, Taiwan (DMR-106-178), China Medical University, Taichung, Taiwan (CMU109-ASIA-06), and Asia University Hospital (10651004 and 10751010). The sponsor or funding organization had no role in the design or conduct of this research.

**Institutional Review Board Statement:** The procedures for animal care and handling were approved by the Animal Committee of China Medical University (permit number: 103-221NH).

**Informed Consent Statement:** Not applicable.

**Data Availability Statement:** All data generated or analyzed during this study are included in this article.

**Acknowledgments:** Experiments and data analysis were performed in part through the use of the Medical Research Core Facilities, Office of Research & Development at China medical University, Taichung, Taiwan.

**Conflicts of Interest:** The authors declare no conflict of interest.

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
