# Peer review of "Anti-Inflammatory Effects of Resveratrol on Human Retinal Pigment Cells and a Myopia Animal Model"

_cimb, doi:10.3390/cimb43020052_

Round 1
Reviewer 1 Report
Review of the manuscript „Anti-Inflammatory Effects of Resveratrol on Human Retinal 2 Pigment Cells and a Myopia Animal Model”
In this manuscript, the authors investigate the effect of resveratrol on myopia by assessing the axial length of the eyes of hamsters with induced myopia, and inflammation markers in ARPE-19 cells upon resveratrol treatment. The paper is well structured and the data are clearly presented. I have some minor comments to improve the manuscript.
Introduction: I suggest to add a sentence about the molecular action of resveratrol and its effect on Sir2
Figures: The bar filling is not consistent. I suggest to stick to one type of bar filling.
Figure 1: Can you include a representative picture of how you measured the axial length in the eye?
Reviewer 2 Report
Manuscript authors ID: cimb-1268053, Title: Anti-inflammatory effects of resveratrol on human retinal pigment cells and a myopia animal model - requires some minor changes.
Introduction: The authors should discuss possible determinants of myopia. In addition, it is advisable that the authors provide a more comprehensive overview of the biological and health role of Resveratrol.
Line 53_ Recent studies have shown that resveratrol has various therapeutic effects in cancers, diabetes, and cardiovascular disease [9]. The authors cite literature [9] from 1976, which is not consistent with the presented disease entities. with therapeutic effects in cancers, diabetes, and cardiovascular disease. R.J.Pryce, P. L., The production of resveratrol by Wis vinifera and other members of the Vitaceae as a response to infection or injury Physiological Plant Pathology 1976, 9 (1), 77-86.
It is advisable that the authors describe the effect of resveratrol in the above-mentioned diseases. The inflammatory and anti-inflammatory effects of myopia were discussed in more detail.
Research methods: A well-prepared research workshop should be emphasized. The authors performed both Cell viability test, enzyme-linked immunosorbent assay, western blotting analysis, etc.
Section: Discussion. The authors described the results of other authors' research more than their own. This needs to be changed. The authors concluded that, "Therefore, these results imply that resveratrol could contribute to the treatment of a variety of inflammatory diseases including myopia." locally or in circulation?
No study limitations.
References: "obsolete" no recent citations.
Author Response
Point 1: Introduction: The authors should discuss possible determinants of myopia. In addition, it is advisable that the authors provide a more comprehensive overview of the biological and health role of Resveratrol.
Response 1: We have rearranged and added sentences to provide more complete background of resveratrol (in lines 54-66).
Recent studies have shown that resveratrol exhibit various therapeutic effects in cancers, diabetes, and cardiovascular diseases. The effects of resveratrol are indicated by decreased serum levels of inflammatory cytokines such as TNF-α, IL-1β, and IL-6 and suppression of the NF-κB pathway for cardiovascular diseases in rats. Resveratrol inhibited inflammasome activation by downregulation of NFκB and P38 mitogen-activated protein kinase (MAPK) expression while upregulation of sirtuin 1 (SIRT1) expression to protect from vascular injury. It also suppressed invasion activity of human hepatocellular carcinoma cell through prohibiting TNF-α mediated MMP9 expression. Resveratrol exhibited anti-obesity effects via down-regulation of adipogenic genes (PPARγ, C/EBPα, FAS, aP2) expression and attenuated the pro-inflammatory cytokines (TNF-α and IL-6) expression. Moreover, resveratrol has been shown as an activator of Sir2 that is a conserved deacetylase for replicative lifespan in aging research. These effects were linked with the anti-inflammatory activity of resveratrol.
Point 2: Line 53_ Recent studies have shown that resveratrol has various therapeutic effects in cancers, diabetes, and cardiovascular disease [9]. The authors cite literature [9] from 1976, which is not consistent with the presented disease entities. with therapeutic effects in cancers, diabetes, and cardiovascular disease. R.J.Pryce, P. L., The production of resveratrol by Wis vinifera and other members of the Vitaceae as a response to infection or injury Physiological Plant Pathology 1976, 9 (1), 77-86.
Response 2: We’ve changed to a more suitable reference which is Acta Biochim Pol, 66 (2019) 13-21.
Point 3: It is advisable that the authors describe the effect of resveratrol in the above-mentioned diseases. The inflammatory and anti-inflammatory effects of myopia were discussed in more detail.
Response 3: We have rearranged and added sentences to provide more complete background of resveratrol (in lines 54-66).
Recent studies have shown that resveratrol exhibit various therapeutic effects in cancers, diabetes, and cardiovascular diseases. The effects of resveratrol are indicated by decreased serum levels of inflammatory cytokines such as TNF-α, IL-1β, and IL-6 and suppression of the NF-κB pathway for cardiovascular diseases in rats. Resveratrol inhibited inflammasome activation by downregulation of NFκB and P38 mitogen-activated protein kinase (MAPK) expression while upregulation of sirtuin 1 (SIRT1) expression to protect from vascular injury. It also suppressed invasion activity of human hepatocellular carcinoma cell through prohibiting TNF-α mediated MMP9 expression. Resveratrol exhibited anti-obesity effects via down-regulation of adipogenic genes (PPARγ, C/EBPα, FAS, aP2) expression and attenuated the pro-inflammatory cytokines (TNF-α and IL-6) expression. Moreover, resveratrol has been shown as an activator of Sir2 that is a conserved deacetylase for replicative lifespan in aging research. These effects were linked with the anti-inflammatory activity of resveratrol.
Point 4: Research methods: A well-prepared research workshop should be emphasized. The authors performed both Cell viability test, enzyme-linked immunosorbent assay, western blotting analysis, etc.
Response 4: The experiments were performed by experienced post-doctors or assistants and further confirmed at least twice to make conclusion.
Point 5: Section: Discussion. The authors described the results of other authors' research more than their own. This needs to be changed. The authors concluded that, "Therefore, these results imply that resveratrol could contribute to the treatment of a variety of inflammatory diseases including myopia." locally or in circulation?
Response 5: We have discussed our results with other researchers at line 297-303 and 370-386.
In the present study, we found resveratrol inhibit myopia progression partly through modulating inflammatory reactions in the eye. We applied the resveratrol directly on to the eye surface which suggested resveratrol acted locally around the eye tissue. However, some studies demonstrated that resveratrol has anti-inflammatory effects on certain diseases via i.v., i.g. or oral administration, such as myocardial ischemia/reperfusion (MI/R) injury or atherosclerosis, which suggested it can also act systemically to lower inflammatory reactions.
- Cong, X.; Li, Y.; Lu, N.; Dai, Y.; Zhang, H.; Zhao, X.; Liu, Y. Resveratrol attenuates the inflammatory reaction induced by ischemia/reperfusion in the rat heart. Mol Med Rep 2014, 9, 2528-2532, doi:10.3892/mmr.2014.2090.
- Deng, Z.Y.; Hu, M.M.; Xin, Y.F.; Gang, C. Resveratrol alleviates vascular inflammatory injury by inhibiting inflammasome activation in rats with hypercholesterolemia and vitamin D2 treatment. Inflamm Res 2015, 64, 321-332, doi:10.1007/s00011-015-0810-4.
Point 6: No study limitations.
Response 6: The limitations of our study is that we used only form-deprivation animal model to induce myopia while not use lens-induced myopia animal model to further confirm the association between inflammation and the pathogenesis of myopia. Moreover, a genetically modified mice or rats with altered inflammatory responses, such as toll-like receptor knockout or knockin animals, would be better to further strengthen our hypothesis, which is that inflammatory reaction is a risk factor for myopia. In addition, we did not determine the exact amount of resveratrol in the eye, which would be helpful in optimizing the dose of the resveratrol used in clinical studies. Currently, we are now determining the amount resveratrol in the eye to optimize the eye drops formulation.
Point 7: References: "obsolete" no recent citations.
Response 7: We’ve modified our citations which added more recent studies.
Here are some of the added references:
- She, M.; Li, B.; Li, T.; Hu, Q.; Zhou, X. Modulation of the ERK1/2-MMP-2 pathway in the sclera of guinea pigs following induction of myopia by flickering light. Exp Ther Med 2021, 21, 371, doi:10.3892/etm.2021.9802.
- Meng, T.; Xiao, D.; Muhammed, A.; Deng, J.; Chen, L.; He, J. Anti-Inflammatory Action and Mechanisms of Resveratrol. Molecules 2021, 26, doi:10.3390/molecules26010229.
- Pyo, I.S.; Yun, S.; Yoon, Y.E.; Choi, J.W.; Lee, S.J. Mechanisms of Aging and the Preventive Effects of Resveratrol on Age-Related Diseases. Molecules 2020, 25, doi:10.3390/molecules25204649.
- Malaguarnera, L. Influence of Resveratrol on the Immune Response. Nutrients 2019, 11, doi:10.3390/nu11050946.
- Li, H.; Xia, N.; Hasselwander, S.; Daiber, A. Resveratrol and Vascular Function. Int J Mol Sci 2019, 20, doi:10.3390/ijms20092155.